# Evaluating the Utility of a Large Language Model in Answering Common Patients’ Gastrointestinal Health-Related Questions: Are We There Yet?

**DOI:** 10.3390/diagnostics13111950

**Published:** 2023-06-02

**Authors:** Adi Lahat, Eyal Shachar, Benjamin Avidan, Benjamin Glicksberg, Eyal Klang

**Affiliations:** 1Chaim Sheba Medical Center, Department of Gastroenterology, Affiliated to Tel Aviv University, Tel Aviv 69978, Israel; 2Mount Sinai Clinical Intelligence Center, Icahn School of Medicine at Mount Sinai, New York, NY 10029, USA; 3The Sami Sagol AI Hub, ARC Innovation Center, Chaim Sheba Medical Center, Affiliated to Tel-Aviv University, Tel Aviv 69978, Israel

**Keywords:** OpenAI’s ChatGPT, chatbot, natural language processing (NLP), medical information, gastroenterology, patients’ questions

## Abstract

Background and aims: Patients frequently have concerns about their disease and find it challenging to obtain accurate Information. OpenAI’s ChatGPT chatbot (ChatGPT) is a new large language model developed to provide answers to a wide range of questions in various fields. Our aim is to evaluate the performance of ChatGPT in answering patients’ questions regarding gastrointestinal health. Methods: To evaluate the performance of ChatGPT in answering patients’ questions, we used a representative sample of 110 real-life questions. The answers provided by ChatGPT were rated in consensus by three experienced gastroenterologists. The accuracy, clarity, and efficacy of the answers provided by ChatGPT were assessed. Results: ChatGPT was able to provide accurate and clear answers to patients’ questions in some cases, but not in others. For questions about treatments, the average accuracy, clarity, and efficacy scores (1 to 5) were 3.9 ± 0.8, 3.9 ± 0.9, and 3.3 ± 0.9, respectively. For symptoms questions, the average accuracy, clarity, and efficacy scores were 3.4 ± 0.8, 3.7 ± 0.7, and 3.2 ± 0.7, respectively. For diagnostic test questions, the average accuracy, clarity, and efficacy scores were 3.7 ± 1.7, 3.7 ± 1.8, and 3.5 ± 1.7, respectively. Conclusions: While ChatGPT has potential as a source of information, further development is needed. The quality of information is contingent upon the quality of the online information provided. These findings may be useful for healthcare providers and patients alike in understanding the capabilities and limitations of ChatGPT.

## 1. Introduction

Gastrointestinal (GI) complaints and symptoms account for approximately 10% of all general practice consultations [1,2] and are apparently very common in the general population. As healthcare providers specializing in this field, we are frequently called upon to answer a wide range of patients’ gastrointestinal health questions. In recent years, large language models, such as OpenAI’s recent release of the ChatGPT chatbot [3], have been developed to provide answers to a wide range of questions in various fields, including healthcare [4].

AI chatbots employ deep learning-based natural language processing (NLP) that evaluates natural human language input and replies accordingly in a conversational mode [5]. These models have the potential to provide patients with quick and easy access to accurate and reliable information about their gastrointestinal health with 24/7 availability, high cost-effectiveness, and potentially less bias based on patients’ demographic characteristics such as gender, race, or age [6]. A recent systematic review assessing the value of AI chatbots in healthcare showed promising results in terms of the effectiveness provided [7].

The recent release of OpenAI’s ChatGPT in November 2022 [3] has garnered immense popularity, as its technical abilities are believed to be superior to previous chatbot versions [8]. In a recent article [8], the chatbot was described as “astonishingly skilled at mimicking authentic writing” and was regarded as “So Good It Can Fool Humans”. It was believed to have passed the ultimate” Turing test”—which states that a machine will be regarded as intelligent if its responses are indistinguishable from those given by a human comparator [9].

The effectiveness of large language models in answering patients’ questions in the field of gastroenterology has not yet been thoroughly evaluated. In this paper, we aim to evaluate the performance of OpenAI’s ChatGPT chatbot in answering patients’ questions concerning gastrointestinal (GI) topics. This evaluation will focus on several key areas, including the accuracy and clarity of the information provided, the ability of the model to handle a wide range of questions, and the overall effectiveness of the model in addressing patients’ concerns and providing them with the information they need to make informed decisions about their health.

Our evaluation aims to provide insights into the capabilities and limitations of large language models in answering patients’ questions in various gastroenterology subjects. Ultimately, our goal is to contribute to the ongoing development of large language models and their use in the field of gastroenterology, with the aim of improving the quality of care and information available to patients.

## 2. Methods

To evaluate the performance of the newly- released large language model, OpenAI’s ChatGPT chatbot, in answering patients’ GI-related questions, we conducted a comprehensive study using a representative sample of real-life questions from patients in this field. The study was designed to assess the accuracy and clarity of the information provided by the chatbot, as well as its overall effectiveness in addressing patients’ concerns and providing them with the information they need to make informed decisions about their health.

The study sample consisted of 110 real-life questions from patients in the field of gastroenterology, gathered from open internet sites providing medical information to diverse patients’ questions. These questions were selected to cover a wide range of topics, including common symptoms, diagnostic tests, and treatments for various gastrointestinal conditions. The questions were selected to reflect the types of questions typically asked by patients in gastroenterology and to provide a representative sample of the types of questions that the chatbot would encounter in a real-world setting.

The questions were provided to the OpenAI chatbot and the answers were recorded. The answers provided by the chatbot were then assessed in consensus by three experienced gastroenterologists, each with more than 20 years of experience. All gastroenterologists work in a tertiary medical center, as well as in community clinics, and together cover all sub-specializations of gastroenterology: IBD experts, motility, hepatology, nutrition, and advanced endoscopy.

The physicians graded each chatbot answer according to a scale of 1–5 (1 the lowest score, 5 the highest) in 4 categories: accuracy, clarity, up-to-date knowledge, and effectiveness in a consensus agreement between all three gastroenterologists. Since the questions were divided according to specific topics—common symptoms, diagnostic tests, and treatments—results are shown in a separate table for every topic. (Table 1—treatments, Table 2—symptoms, Table 3—diagnostic tests.) Results are summarized visually in Figure 1, Figure 2 and Figure 3, respectively. All questions and chatbot answers are attached in the Appendix A.

### Statistical Analysis

Python (Ver. 3.9) was used for all statistical calculations. The average score ± standard deviation (SD) was calculated for each category and is shown in a table. A distribution graph of the answer grades for all groups by answer category was plotted. A Kruskal–Wallis one-way analysis of variance was used to determine the statistical difference between the groups.

## 3. Results

Overall, 110 various typical patient questions were presented to the OpenAI chatbot. Forty-two included various questions regarding treatment options in diverse gastroenterology fields. Questions were chosen to include the whole spectrum of specification from very specific questions (e.g., treatment for GI cancers according to the stage) to wider questions not referring to definite disease staging or severity (e.g., “what is the treatment for ulcerative colitis”?). All questions chosen were short and clear for better standardization. The results of the evaluation for the treatment questions are shown in Table 1.

A full list of all questions and answers is presented in the Appendix A.

Overall, the OpenAI chatbot answers to the questions dealing with treatment options varied wildly between the different topics and were inconsistent in their level. Notably, at the end of almost all the answers, the patient is referred to a doctor/healthcare provider for further evaluation.

The results of the evaluation for questions related to symptoms are shown in Table 2. In the answers to the symptom-related questions, the OpenAI chatbot shows the same pattern, and the majority of answers referred the patient to consult a physician.

The results of the evaluation for questions related to diagnostic examinations are shown in Table 3.

Diversity in the quality of answers to the diagnostic examination was the highest of all groups. While many answers were correct and informative, other medical terms were simply not recognized by the chatbot (Table 3).

Figure 1, Figure 2 and Figure 3 graphically display the distribution pattern of grades assigned to the responses provided by the OpenAI chatbot in relation to treatment, symptoms, and diagnostic-related questions, respectively.

As shown graphically in Figure 1, Figure 2 and Figure 3, the distribution of grades by answer group in the treatment-related and symptoms-related groups are similar and relatively consistent, while the distribution of the diagnostic-related group is wide and inconsistent.

Statistical difference between the groups was shown for the subjects of accuracy (*p* = 0.019) and up-to-date (*p* = 0.009), but not for the subjects of clarity (*p* = 0.071) and efficacy (*p* = 0.071).

## 4. Discussion

Overall, our study provides a comprehensive evaluation of the performance of a large language model, OpenAI’s ChatGPT chatbot, in answering patients’ questions in the field of gastroenterology. The results of the study will hopefully be valuable for healthcare providers and patients alike, as they will provide important insights into the capabilities and limitations of these models in providing accurate and reliable information about gastrointestinal health.

Generally, the results of our evaluation of OpenAI’s ChatGPT large language model, as a potential tool for answering patients’ GI-related questions, are concerning. Despite the impressive capabilities of this model in other applications, our study found that it was insufficient for reliably answering patients’ questions in this domain.

In our study, we presented the model with a range of common gastroenterological questions and assessed its ability to provide accurate and helpful responses. While the model could generate responses for all questions, the accuracy and helpfulness of these responses varied significantly. In many cases, the model’s responses were either incomplete or entirely incorrect, indicating a lack of understanding of the subject matter.

These findings are particularly concerning given the importance of accurate information in the medical field. Patients often rely on Internet information to make decisions about their health and well-being, and inaccurate information can have serious consequences. As such, it is essential that any tool used to provide medical information to patients be able to provide accurate and reliable responses.

Interestingly, while the answers to treatment-based questions and symptom-based questions showed a similar and relatively steady quality and effectivity (Table 1 and Table 2, Figure 1), diagnostic-related results diverged between high performance to incorrect/missing data (Table 3, Figure 1). These results may improve with time as the OpenAI chatbot acquires more data.

Furthermore, our study also highlighted the limitations of OpenAI’s ChatGPT chatbot in the medical domain. The ability to understand and accurately respond to complex medical questions requires a deep understanding of the subject matter, as well as the ability to process and integrate information from a variety of sources [10]. While ChatGPT is an impressive model with many capabilities, it is not yet advanced enough to fulfill this role. Notably, it should be acknowledged that the utilization of real-world questions in our study resulted in the inclusion of numerous acronyms, which could potentially pose challenges in web-based research settings. However, due to our intention to assess the effectiveness of ChatGPT in addressing authentic patient inquiries, we refrained from including the complete expressions, despite the potential difficulties associated with the acronyms.

Our results are in line with previous data assessing other artificial intelligence conversational agents in healthcare [6,7,11]. As a group, the bot’s function was found to be relatively satisfactory and useful, with moderate effectiveness. It is, therefore, important to characterize specific strengths and weaknesses of the chatbot in order to maximize its benefits.

Recent data from the last months suggest that LLMs hold the potential for significant impacts in medicine [12,13].

In our previous work, we examined the utility of ChatGPT in identifying top research questions in gastroenterology. Our conclusion was that ChatGPT may be a useful tool for identifying research priorities in the field of GI, yet we noted the need to improve the originality of the research questions identified [12].

In telehealth, LLMs such as ChatGPT could improve efficiency and scalability, and provide unbiased care [14]. Despite its proficiency in generating differential diagnosis lists, ChatGPT does not surpass human physicians [15]. Moreover, LLMs can offer innovative methods in medical education [16]. Yet, it is crucial to address LLM’s limitations to maintain accuracy and reliability in patient care. Furthermore, Rasmussen et al. [17] and Samaan et al. [18] evaluated the accuracy of ChatGPT in generating responses to typical questions related to vernal keratoconjunctivitis and bariatric surgery, respectively. Both found the model to be broadly accurate. Xie et al. [19] conducted a similar study using a simulated rhinoplasty consultation, and while they observed that ChatGPT provided easily comprehensible answers, it was less effective at providing detailed, personalized advice. Yeo et al. [20] tested ChatGPT on questions about cirrhosis and hepatocellular carcinoma, finding the AI capable but not comprehensive in its knowledge. Johnson et al. [21] investigated the accuracy of ChatGPT in generating responses to common cancer myths and misconceptions, concluding that it often provided accurate information. Lastly, a comprehensive pilot study on ChatGPT by Johnson et al. [22] discovered that its responses were largely accurate across a wide range of medical questions.

Significantly, it is worth mentioning that all of these studies have been published within the recent months subsequent to the initial release of the inaugural version of ChatGPT. This accumulation of research findings attests to the remarkable potential impact and significance that ChatGPT holds for the field of medicine in the foreseeable future.

Our study had a few limitations. First, the choice of questions naturally influenced the answers obtained by the OpenAI chatbot. However, the wide variety of questions and their relatively high number minimized the chance for bias. Furthermore, the answers were graded with the consensus of three physicians; there is a possibility that other physicians might have had a different point of view. Nevertheless, to minimize that risk, we chose highly experienced gastroenterologists with wide clinical experience and more than 70 years of practicing gastroenterology both in a tertiary medical center and in community practice. Lastly, there were no comparisons with other chatbots to gauge the relative merits of ChatGPT against other similar technologies, and no blinded comparison between human and chatbot answers was performed. These research directions necessitate further dedicated exploration to explore and investigate its potential in greater detail.

Importantly, our study did not include patients’ comments and, therefore, is not valid in terms of assessing patients’ points of view. This merits further specific research.

Overall, our study suggests that caution should be exercised when considering the use of a large language model, such as OpenAI’s ChatGPT, for providing medical information to patients. While the model has the potential to be a powerful tool, it is not yet ready to be relied upon for this critical task. Further research and development are needed to improve its capabilities in the medical domain before it can be used with confidence.

## Figures and Tables

**Figure 1 diagnostics-13-01950-f001:**
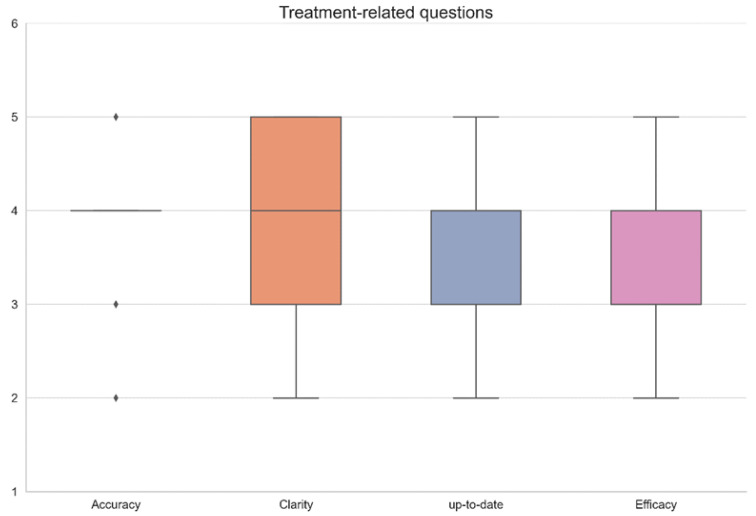
Graphic distribution pattern of grades assigned to the responses provided by the OpenAI chatbot in relation to treatment-related questions.

**Figure 2 diagnostics-13-01950-f002:**
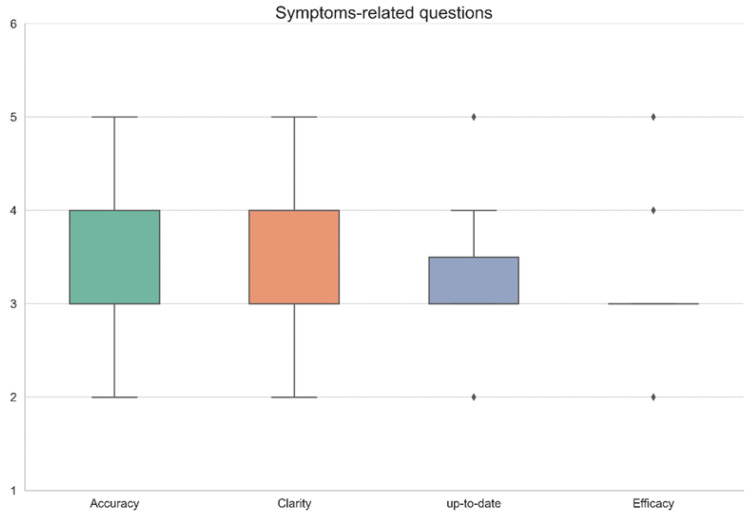
Graphic distribution pattern of grades assigned to the responses provided by the OpenAI chatbot in relation to symptoms-related questions.

**Figure 3 diagnostics-13-01950-f003:**
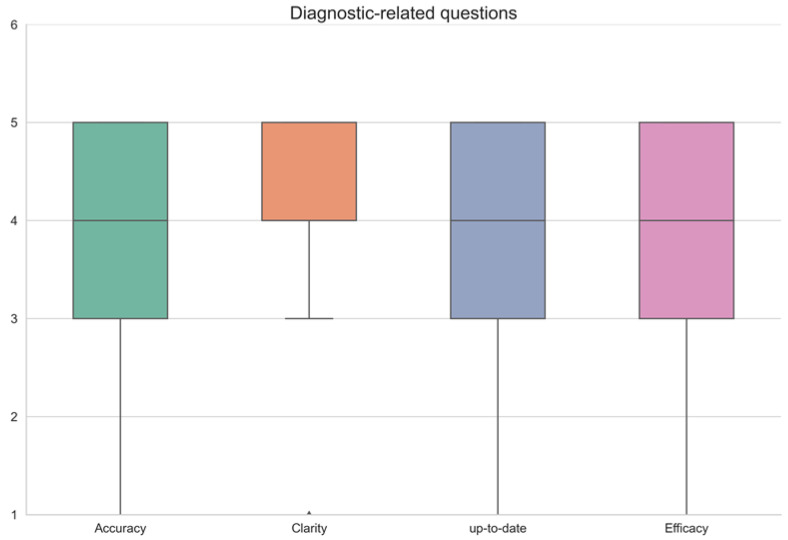
Graphic distribution pattern of grades assigned to the responses provided by the OpenAI chatbot in relation to diagnostic-related questions.

**Table 1 diagnostics-13-01950-t001:** Grades of the OpenAI chatbot answers to treatment-related questions.

Question No.	Accuracy	Clarity	Up-to-Date	Efficacy	Remarks
1	3	2	3	3	
2	3	4	3	3	
3	4	3	4	3	
4	4	4	4	3	
5	4	4	3	2	
6	5	5	4	4	
7	4	4	3	2	
8	4	4	3	2	
9	4	3	4	3	
10	5	5	4	4	
11	4	4	4	3	
12	4	4	3	3	
13	3	3	3	2	
14	4	3	4	3	
15	2	4	2	2	
16	4	3	4	4	
17	4	3	4	3	
18	4	5	5	4	
19	3	3	2	2	
20	3	4	4	3	
21	2	3	2	2	
22	4	4	4	5	
23	4	2	3	2	
24	3	4	3	3	
25	4	4	5	4	
26	4	5	5	4	
27	5	5	4	4	
28	4	5	5	4	
29	4	5	4	4	
30	5	4	5	5	
31	4	5	4	4	
32	4	5	4	3	
33	5	5	5	5	
34	4	3	4	3	
35	5	4	4	4	
36	3	3	4	3	
37	2	3	2	2	
38	4	3	4	3	
39	5	4	5	5	
40	4	5	4	4	
41	5	5	5	5	
42	4	4	5	4	
Average ± SD	3.9 ± 0.8	3.9 ± 0.9	3.8 ± 0.9	3.3 ± 0.9	
Median (IQR)	4 (4–4)	4 (3–5)	4 (3–4)	3 (3–4)	

**Table 2 diagnostics-13-01950-t002:** Grades of the OpenAI chatbot answers to symptoms-related questions.

Question No.	Accuracy	Clarity	Up-to-Date	Efficacy	Remarks
1	3	4	3	3	
2	5	5	5	5	
3	3	4	3	3	
4	3	4	3	3	
5	3	4	3	3	
6	3	3	3	3	
7	4	4	4	4	
8	3	4	3	3	
9	3	4	3	3	
10	4	4	4	4	
11	3	3	3	3	
12	3	3	3	3	
13	2	3	2	2	
14	3	2	3	2	
15	4	4	4	3	
16	3	4	3	3	
17	3	4	3	3	
18	4	4	3	3	
19	3	3	3	3	
20	3	4	3	3	
21	5	3	4	4	
22	3	4	3	3	
23	5	5	5	5	
Average ± SD	3.4 ± 0.8	3.7 ±0.7	3.3 ± 0.7	3.2 ± 0.7	
Median (IQR)	3 (3–4)	4 (3–4)	3 (3–3.5)	3 (3–3)	

**Table 3 diagnostics-13-01950-t003:** Grades of the OpenAI chatbot answers to diagnostic-related questions.

Question No.	Accuracy	Clarity	Up-to-Date	Efficacy	Remarks
1	5	5	5	5	
2	5	4	4	3	
3	4	5	5	5	
4	4	5	5	4	
5	5	4	5	4	
6	5	4	4	5	
7	0	0	0	0	Not found
8	5	4	5	4	
9	1	1	1	1	Mistake
10	5	5	5	5	
11	5	5	5	5	
12	4	5	4	4	
13	5	5	5	5	
14	5	5	5	4	
15	5	5	5	5	
16	5	4	5	4	
17	5	4	5	4	
18	4	4	5	4	
19	5	4	5	4	
20	4	5	4	4	
21	5	3	3	3	Asks info
22	5	5	5	5	
23	0	0	0	0	Not found
24	5	5	5	5	
25	3	4	4	3	
26	3	4	4	3	
27	4	4	4	4	
28	5	5	5	5	
29	5	5	5	5	
30	3	4	3	3	
31	5	5	5	5	
32	4	5	5	4	
33	3	4	4	3	
34	5	5	4	5	
35	5	5	5	5	
36	1	1	1	1	Mistake
37	0	0	0	0	Not found
38	0	0	0	0	Not found
39	3	4	3	3	
40	4	5	4	4	
41	0	0	0	0	Not found
42	0	0	0	0	Not found
43	3	5	0	2	
44	5	5	4	5	
45	4	5	4	4	
Average ± SD	3.7 ± 1.7	3.8 ± 1.7	3.6 ± 1.8	3.5 ± 1.7	
Median (IQR)	4 (3–5)	4 (4–5)	4 (3–5)	4 (3–5)	

## Data Availability

Due to data privacy regulations, the raw data of this study cannot be shared.

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
