# Peer review of "Evaluating the Utility of a Large Language Model in Answering Common Patients’ Gastrointestinal Health-Related Questions: Are We There Yet?"

_diagnostics, 2023, doi:10.3390/diagnostics13111950_

Round 1

Reviewer 1 Report

An emerging concept and the authors have done well to conduct this study. 

  1. The study does add to the novelty of this emerging concept of AI   
  2. This study and further studies in this domain is likely to generate reader interest provided tangible data with clinical relevance is obtained in the future 

The following maybe discussed to enhance the discussion:

a. Lahat A, Shachar E, Avidan B, Shatz Z, Glicksberg BS, Klang E. Evaluating the use of large language model in identifying top research questions in gastroenterology. Sci Rep. 2023;13(1):4164. Published 2023 Mar 13. doi:10.1038/s41598-023-31412-2

b. Ge J, Lai JC. Artificial intelligence-based text generators in hepatology: ChatGPT is just the beginning. Hepatol Commun. 2023 Mar 24;7(4):e0097. doi: 10.1097/HC9.0000000000000097. PMID: 36972383; PMCID: PMC10043591.

c. Hirosawa T, Harada Y, Yokose M, Sakamoto T, Kawamura R, Shimizu T. Diagnostic Accuracy of Differential-Diagnosis Lists Generated by Generative Pretrained Transformer 3 Chatbot for Clinical Vignettes with Common Chief Complaints: A Pilot Study. Int J Environ Res Public Health. 2023 Feb 15;20(4):3378. doi: 10.3390/ijerph20043378. PMID: 36834073; PMCID: PMC9967747.

d. Eysenbach G. The Role of ChatGPT, Generative Language Models, and Artificial Intelligence in Medical Education: A Conversation With ChatGPT and a Call for Papers. JMIR Med Educ. 2023 Mar 6;9:e46885. doi: 10.2196/46885. PMID: 36863937; PMCID: PMC10028514.

Reviewer 2 Report

This is an interesting study evaluating the use of the Open AI’s ChatGPT chat bot in answering common questions in the area of gastroenterology and hepatology. Given the current interest and development of artificial intelligence and chatbots, this is an interesting topic.

The abstract is complete with results provided clearly. The methods would benefit from clarification of the process of comparison of the answers by ChatGPT and the physicians. It appears the physicians rated the chat bot answers and they were not actually compared to answers that the physicians might have given. This is misleading and would suggest a different method than what was used. The conclusion is that further development of the chat bot is needed, however there should also be consideration of the material available on-line as that is the source of the information. If information on-line is incomplete, answers will be incomplete.

The background is sufficient with information about AI chat bots and possible use by patients in answering questions related to gastrointestinal questions.

The methods section has a good description of the questions identified and the use of the AI chat bot to obtain answers. The evaluation component of the study is not clear. It states that the answers provided by the AI chat bot were compared to those of 3 gastroenterologists, however the gastroenterologists rated the answers. They did not provide answers that were then compared by other parties. The gastroenterologists rated the answers in those areas for accuracy, clarity, up-to-date information, and effectivity. It is not clear what is meant the effectivity. Do you mean effectiveness? This should be defined as to what was meant. The methods note that all 3 gastroenterologists rated the answers and grades were summarized, but it also says consensus was reached. Did they rate independently or reached consensus? The answers would appear to indicate consensus as there is a single, whole number for each item. Please clarify the process and rating used.

The data are presented clearly and it was nice that the questions and answers were provided in the supplemental material.

The results section has some of the methods repeated or added. There are some comments in the results that indicate opinion rather than an objective assessment of the results. For example, on page 5 authors note that answers related to treatment options varied wildly between topics. It would be better to state the variance rather than use a subjective term.

The discussion is fairly complete. One factor that is not considered is the source of the information for the AI chat bot which is electronic sources. Results would also indicate that detailed information related to the questions is missing from electronic sources. This would be consistent with what most gastroenterology departments or health care institutions post related to these topics. Research would not include answers to these basic questions. The use of some of the acronyms in the questions should be addressed as students are aware that use of acronyms in web-based searches is problematic so why were those included as such and not spelled out? This should also be noted in the discussion.

Limitations are clearly addressed.

The conclusion is that the AI chat bot should be used with caution. It would be important to note the source of information used as well.

Reviewer 3 Report

Review: „Evaluating the Utility of a Large Language Model in Answering  Common Patients’ Gastrointestinal Health-Related Questions- Are We There Yet?“

General comments

What was the research question?

With the emergence of large language models (chatbots) and the tendency of patients to first look for answers to health-related questions in search machines rather than in books or to ask health professionals directly it is important that chatbots provide accurat, clear and up-to-date information. The article tries to answer the question whether OpenAI’s ChatGPT in accurate und up-to-date in answering patients’ questions about gastrointestinal health.

Research method:

110 Questions were grouped into the categories: questions regarding symptoms, diagnosis, and treatment

A panel of 3 Gastroenterologists graded the answers

Measured variables: Accuracy, Clarity, Ability to handle a wide range of questions, Overall effectiveness of model in addressing patients‘ concerns and providing information for an informed health care decision

Q1: How are accuracy, clarity, up-to-date and effectiveness defined by the panel of gastroenterologists. Is the grading based on documented scheme? How can these results reproduced?

Controls: No controls.

Q2: It would be interesting how a general practice provider or gastroenterologist is graded by the panel of experts to give a better understanding how well ChatGPT performs. In this scenario the panel is blinded to the origin of the answer. Then a real baseline for accuracy, clarity and ability to answer questions is established.

Detailed comments:

Q3: Methods:

1.      Statistical analysis was based on the non-parametric Kruskal-Wallis-Test. At the same time average and standard deviation are reported in a parametric manner. Those two methods do not fit together. Please also provide Median and Min, Max, Interquartile range.

Figure 1 one gives an indication that the data is not normal-distributed, which makes the average and subsequently the standard deviation biased.

Q4 Results:

1.       Table 1 to 3 give the grades to treatment / symptoms / diagnostic related questions as they were graded by the experts. These tables present no summarisation of the data. Please consider graphs, i.e. boxplots in addition to the average and standard deviation to summarise the data and facilitate the results.

2.       What does Figure 1 actually show? What is the purpose of this figure?
Are now grades for all four measured categories (accuracy, up-to-date, etc.) for all questions in each category (treatment, symptoms, and diagnosics) aggregated? To which grade are the authors referring? Or is it an overall grade? Please label the x-axis acordingly. Maybe a histogram based figure is here more informative than one based on density plots.

3.       Results are describesd as informative, shallow, worrisome, helpful, stereotypical etc. how is this measured? I would say it belongs in the discussion not the results.
The authors wanted to describe accuracy, clarity, if they were up-to-date and efficacy (as stated in the tables) or effectiveness (as stated in the introduction) of the model in answering patients question.

Q5 Discussion:

1.       Line 199 ff: the authors state that there is no literature assessing the capabilities of chatGPT in the medical field. A short pubmed search yield a diverse list of literature addressing exactly that. I would suggest the renew the literature search and include current literature.

The newest literature cited is from December 2022. In the time until now a few articles were published on the accuracy and reliability of ChatGPT on patients‘ questions regarding different fields of medicine (PMIDs: 371598147 (breast augmentation), 37129631 (vernal keratoconjunctivitis), 37106269 (bariatric surgery), 37095384 (rhinoplasty), 36946005 (cirrhosis and heptocellular carcinoma), 36929393 (cancer), 36909565 (diverse), 36834073 (pilot study on differential diagnoses)).

Q6: Institutional Review Board Statement is incomplete

Q7: Informed Consent Statement is incomplete

Round 2

Reviewer 3 Report

Dear authors,

Thank you for the article and your work.

My comments have been addressed sufficiently and I'm happy to support the publication.